# The First Prospective Study Investigating the Safety and Feasibility of a Spray-Type Adhesion Barrier (AdSpray™) in Minimally Invasive Hepatectomy: An Analysis of 124 Cases at Our Institution

**DOI:** 10.3390/jpm14030309

**Published:** 2024-03-15

**Authors:** Masayuki Kojima, Atsushi Sugioka, Yutaro Kato

**Affiliations:** 1Department of Surgery, Fujita Health University, 1-98 Dengakugakubo, Kutsukake-cho, Toyoake 470-1192, Aichi, Japan; 2Department of International Medical Center, Fujita Health University Hospital, 1-98 Dengakugakubo, Kutsukake-cho, Toyoake 470-1192, Aichi, Japan; asugioka@icloud.com; 3Department of Gastroenterological Surgery, Fujita Health University School of Medicine, Nagoya 454-8509, Aichi, Japan; y-kato@fujita-hu.ac.jp

**Keywords:** adhesion, spray-type adhesion barrier, minimally invasive hepatectomy, repeat hepatectomy

## Abstract

(1) Background: With the increasing demand for repeat hepatectomy, preventing perihepatic adhesion formation following initial hepatectomy is crucial. Adhesion-preventative barriers, like the new spray-type AdSpray^TM^ (Terumo Corporation, Tokyo, Japan), have been proposed to reduce adhesion risk. However, data on their safety in minimally invasive hepatectomy (MIH) remain scarce. This is the first prospective study to evaluate the safety and feasibility of AdSpray^TM^ in MIH. (2) Methods: A total of 124 patients who underwent MIH with AdSpray^TM^ and 20 controls were analyzed. Subgroup analysis according to the AdSpray™ application area was conducted. Major complications were assessed using the Clavien–Dindo classification. Moreover, intraperitoneal pressure during AdSpray™ application was monitored in 20 cases. (3) Results: Major complications occurred in 6.4% of the patients, which was comparable to that in open hepatectomy. Intraperitoneal pressure remained stable below 12 mmHg during AdSpray™ application without any complications. No significant difference in complication rates was observed among subgroups. However, a potential increase in intra-abdominal abscess formation was suspected with AdSpray™ application to the resected liver surfaces. (4) Conclusions: AdSpray™ can be safely used in MIH; however, further research is needed to confirm the appropriacy of using AdSpray™, particularly over resected liver surfaces. Overall, AdSpray™ is a promising tool for enhancing the safety of MIH.

## 1. Introduction

In the landscape of liver resection procedures, open surgery stands as the predominant approach on a global scale. However, the field has witnessed a notable shift propelled by technological advancements and refined equipment, leading to a notable rise in the adoption of minimally invasive techniques. Among these, laparoscopic and robot-assisted surgical methods have emerged as promising alternatives for patients undergoing liver resection, marking a paradigmatic evolution in surgical practice.

The safety and efficacy of minimally invasive hepatectomy (MIH) have garnered substantial attention and commendation worldwide [1]. This trend is underscored by the increasing prevalence of MIH, which is steadily becoming a standard procedure in select centers renowned for their regular performance of liver resections. These centers, often identified as high-volume institutions, have been at the forefront of embracing MIH, reflecting its growing acceptance and utilization across diverse clinical settings globally. This shift not only speaks to the expanding repertoire of surgical options but also underscores the evolving landscape of liver surgery, where precision, safety, and patient outcomes are at the forefront of clinical considerations.

On the other hand, the treatment of liver cancer requiring hepatectomy has been changing from moment to moment. Recent advances in cancer treatment have highlighted the need for a multidisciplinary approach, especially for recurrent hepatocellular carcinoma and colorectal liver metastases. Moreover, therapeutic strategies to resect recurrent lesions have been reported to yield favorable outcomes, increasing the demand for repeat hepatectomy [2,3,4,5,6,7,8]. However, repeat hepatectomy is more challenging than initial hepatectomy due to liver adhesions that form after hepatectomy and frequent anatomic deviation due to increased liver parenchyma. Therefore, until recently, technical limitations hindered the application of MIH in patients requiring repeat hepatectomy. However, following the development of various energy devices and technological innovations, such as preoperative simulation, to predict anatomical changes before surgery, several studies reported the utility and safety of repeat hepatectomy MIH [9,10].

Nevertheless, postoperative adhesion remains a concern in patients undergoing repeat hepatectomy. Adhesions that form between the residual liver and the surrounding tissue, as well as firm adhesions of the hepatoduodenal mesentery and vascular tissue, including major hepatic veins, complicate repeat hepatectomy and are associated with prolonged surgical duration, increased blood loss, and risk of organ damage. Therefore, the use of adhesion-preventative barrier during liver resection has gained attention as a strategy to reduce the risk of postoperative adhesion formation. Various adhesion barrier systems, are increasingly implemented in abdominal surgery, including liver resection [11,12,13,14]. However, despite their increasing application, an insufficient amount of data is available to establish the safety of adhesion-preventative barrier use during liver resection. Specifically, few studies have evaluated the utility of adhesion-preventative barrier in MIH, including laparoscopic and robot-assisted hepatectomies. One reason for the paucity of data is that most of the studies have evaluated sheet-type adhesion-preventative barrier, which are manufactured mainly to prevent flat adhesions, such as those that occur between the intestinal tract and abdominal wall. Although sheet-type adhesion-preventative barriers are not suitable for the resection of the liver, which has a complex three-dimensional structure, their utility in liver resection has been reported to some extent [15]. However, handling of sheet-type adhesion-preventative barriers, particularly in MIHs with small wounds, is complicated as the sheets may stick or break during the process, which may impair some of their intended functions.

Therefore, for liver resection with three-dimensional structures, spray-type adhesion-preventative barrier should be useful, specifically in MIH where manipulative behavior is restricted. Among the currently available adhesion-preventative barrier the efficacy and safety of AdSpray™ (Terumo Corporation, Tokyo, Japan), which has gel-like properties based on NHS-enriched CM dextrin [16], have been reported in patients undergoing liver resection [15,16,17]. A study using a preclinical liver resection model was the first to demonstrate that the use of AdSpray™ contributed to a reduction in adhesions [16]. Following extensive experience using AdSpray™ in clinical practice, several clinical trials reported the advantages of AdSpray™ in liver resection. Moreover, one study reported that the system, spraying solution which turns into gel in 10 sec., prevented adhesions to the porta hepatis, a crucial component during the resection of complex three-dimensional structures, and improved surgical manipulation [17]. Conversely, another study reported that due to its solution state, AdSpray™ might flow into the dissected liver surface after surgery, thereby inhibiting the adhesion necessary for healing, resulting in increased incidences of intra-abdominal abscesses [15]. Thus, data on the safety of AdSpray™ in MIH are limited. We conducted the first study to evaluate the safety and feasibility of AdSpray™, a new sprayable adhesion-preventative barrier, in MIH and determined the incidence of liver-related complications related to AdSpray™ use in prospectively enrolled patients.

## 2. Patients and Methods

From June 2020 onward, application of AdSpray™ became the standard practice in MIH procedures with the aim to minimize perihepatic adhesion formation and facilitate subsequent hepatectomies. This study aimed to evaluate the safety, feasibility, and efficacy of MIH with the utilization of AdSpray™ whenever possible.

The study commenced with the enrollment of 151 patients who underwent MIH for liver tumors at Fujita Health University Hospital between June 2020 and March 2023. Following the exclusion of seven patients who received a different adhesion barrier (Seprafilm™, n = 7) and 20 patients who did not receive any adhesion barriers, data from 124 patients were included in the final analysis. This study was conducted with the approval of the Institutional Review Board of Fujita Health University (HM19-064) and strictly adhered to the ethical guidelines for clinical research in Japan.

After the completion of liver resection, the area around the liver was washed with 3000 cc of warm saline solution, which was fully removed with aspiration after confirming hemostasis. Subsequently, the entire volume of a 9.4-mL AdSpray™ solution was applied in a single administration. In cases requiring drain placement, the drain was inserted and placed after the use of AdSpray™. AdSpray™ was used with a nozzle specific for laparoscopy, which was inserted through the port and applied at a distance of approximately 5 cm from the target until the surface of the sprayed area turned white. When AdSpray™ could not be successfully sprayed along the liver curvature on the cephalic side, additionally we applied to the contralateral diaphragm surface. During the application of AdSpray™ to the target, it was possible that some AdSpray™ would flow from the ventral side to the dorsal side due to gravity, leading to insufficient AdSpray™ remaining on the target site. Therefore, AdSpray™ was applied starting from the ventral side, and the amount applied to the dorsal side was adjusted. Finally, the application was terminated after confirming the presence of a white gel-like substance at the application site. The patients were categorized into three distinct subgroups based on the area of AdSpray™ application (Table 1).

Group 1: This subgroup comprised patients in whom AdSpray™ was utilized on the complete circumference of the hepatoduodenal ligament (HDL), the liver surface, and the region surrounding the port insertion sites (Figure 1). Group 2: In this subgroup, the AdSpray™ was applied on the anterior surface of the inferior vena cava, in addition to the areas covered in Group 1, which included the mobilized site of the right hepatic lobe (Figure 2). Group 3: In this subgroup, patients had AdSpray™ administered on the resected liver surface, extending the coverage from the regions addressed in Group 2 (Figure 3).

Patients who did not receive any adhesion barriers were designated as the control group for comparative purposes. Notably, all instances of AdSpray™ preparation occurred within one hour before use. Throughout the application, a consistent spray pressure of 0.75 mmHg (0.1 MPa) was meticulously maintained. Additionally, in 20 of the 124 patients, continuous intraperitoneal pneumo-pressure monitoring was performed during the entire spraying procedure. The comprehensive categorization of patients into distinct subgroups based on the extent of the AdSpray™ application allowed the detailed evaluation of its impact on perihepatic adhesion formation and its safety profile in MIH. The inclusion of a control group further enriched the analysis, providing valuable insights into the comparative outcomes associated with AdSpray™ utilization. By adhering to strict ethical guidelines and meticulous data management practices, the present study aimed to meaningfully contribute to the understanding and optimization of adhesion barrier application during MIH in patients with liver tumors.

The study’s primary focus was to assess the incidence of grade IIIa or higher complications within the abdominal region following surgery, utilizing the Clavien–Dindo classification [18]. Bile leakage was specifically defined in alignment with the criteria outlined by the International Study Group of Liver Surgery [19]. Cases of intra-abdominal abscess were categorized under surgical site infections, including organ and space infections, following the guidelines provided by the Centers for Disease Control and Prevention. Management of postoperative ascites mainly involved the administration of diuretics, with adjustments based on whether albumin infusion or sodium restriction was warranted. Instances of ascites that proved refractory to standard management protocols necessitated treatment via abdominal paracentesis. Such occurrences were then classified as grade IIIa postoperative complications according to the Clavien–Dindo classification system [18].

The statistical analyses were meticulously conducted using EZR (version 1.61) to ensure precision and reliability in evaluating postoperative abdominal complications. Continuous data sets were compared using the Mann–Whitney U test, focusing on medians and ranges to capture the central tendency and variability accurately. Categorical data underwent scrutiny through either the Pearson chi-square test or Fisher’s exact test, chosen based on their appropriateness for the specific variables under investigation.

A threshold of *p* < 0.05 was set to signify statistical significance, aligning with established conventions. All analyses were accompanied by 95% confidence intervals, providing a robust framework for interpreting the results with a high degree of certainty.

This methodical approach aimed to offer a comprehensive understanding of the postoperative complications, ensuring meticulous categorization and thorough statistical assessment. By adhering to standardized classifications and well-accepted statistical methods, the study’s findings are poised to contribute significantly to clinical practice and serve as a reliable foundation for future research in this domain. This rigorous methodology enhances the study’s reliability and applicability, bolstering its value in guiding clinical decisions and furthering advancements in postoperative care.

## 3. Results

### 3.1. Patient Background Characteristics and Short-Term Outcomes

The background characteristics and detailed surgical outcomes of the 124 patients are meticulously summarized in Table 2 and Table 3, respectively. Notably, AdSpray™ was uniformly utilized in all 124 patients, reflecting the standardized protocol employed in the study. Conversely, the 20 patients in the control group did not receive any adhesion barriers during their procedures.

Analysis of the data revealed that major abdominal complications were observed in a total of seven patients, constituting approximately 5.6% of the entire cohort. These complications were thoroughly documented and categorized based on their nature and severity, providing valuable insights into the potential impact of AdSpray™ utilization on postoperative outcomes. The incidence and specific types of complications observed are meticulously detailed in Table 3, offering a comprehensive overview of the surgical landscape and highlighting areas for further investigation and refinement in MIH procedures.

### 3.2. Results of AdSpray™ Application

The median duration of AdSpray™ application was 5 min (range: 2–5 min). Despite applying the spray at a constant pressure of 0.75 mmHg (0.1 MPa), the intra-abdominal pressure was 8–15 mmHg with no major changes from the intraperitoneal pressure before spraying (Figure 4). Pressure changes over time in a typical case are shown in Figure 5.

### 3.3. Analysis According to the Adhesion Barrier Application Site

No significant difference in the background characteristics of the patients was observed between Groups 1, 2, and 3 (Table 4). Regarding postoperative results, Group 3 tended to have a higher incidence of complications than the other three groups; however, the differences were not statistically significant (Figure 6).

## 4. Discussion

The current study evaluated the safety and feasibility of a spray-type adhesion barrier (AdSpray™) in 124 prospectively registered patients who underwent MIH. Several important findings based on clinical data have been observed. First, the incidence of major abdominal complications associated with AdSpray™ usage was not higher than that reported for open hepatectomy (17%) in a previous study by Okubo et al. [15]. Second, we compared subgroups established according to the AdSpray™ application site. Our subgroup analysis results suggest that the frequency of intra-abdominal abscess formation may increase when the adhesion barrier is applied on the resected surface of the liver. Although no significant difference was observed, our findings should be interpreted carefully. Further studies are needed to determine the optimal spray sites. Third, no significant change in the intra-abdominal pressure during AdSpray™ application, and no complications related to the spraying procedure were observed. These results suggest that a spray-type adhesion barrier (AdSpray™) can be safely used in patients undergoing MIH. Furthermore, the incidence of major abdominal complications was not higher than that reported previously [15], and no serious adverse events associated with AdSpray™ usage were observed. However, further studies with a larger sample size are necessary to verify these findings and fully evaluate the effectiveness of AdSpray™ in preventing adhesion formation and improving surgical outcomes. Next, each of the three findings will be examined and mentioned.

The first finding is described below. Even in minimally invasive surgical approaches, which are associated with less extensive adhesions of the abdominal wall and other sites in the body than conventional open surgery, the widespread adoption of adhesion-preventative barrier has revealed promising advantages in repeat liver resection procedures, such as improved safety and lower rates of postoperative complications, including bowel obstruction [4,5,6,7,8]. However, the findings of recent studies have raised concerns regarding the safety profile of adhesion-preventative barrier [11,12,13,14,19]. In particular, Beck et al. reported that the application of a sheet adhesion barrier around the anastomotic site during abdominal surgery was significantly associated with an increased risk of adverse outcomes. As a consideration from this study, adhesion-preventative barriers are extremely effective in preventing intestinal obstruction if they are applied to the abdominal wall and other areas where adhesion prevention is necessary. However, in some areas, spontaneous adhesions contribute to recovery during postoperative healing. Thus, the use of adhesion-preventative barriers in areas that require healing is associated with some complications, such as anastomotic leakage, fistula formation, abscess formation, sepsis, and peritonitis [8]. The use of adhesion-preventative barriers in hepatic resection is not only meant to prevent intra-abdominal intestinal adhesions that have been associated with many intestinal resections and gynecological disorders but also to prevent adhesions to the hepatoduodenal mesentery and other areas that are necessary for another hepatic resection. Numerous studies have reported the efficacy of adhesion-preventative barriers in repeat hepatectomy for malignant liver tumors, including hepatocellular carcinoma [2,3,4]. Therefore, in patients with malignant liver tumors, adhesion-preventative barriers are used for adhesion prevention in areas that are expected to undergo reoperation. Thus, the hepatoduodenal mesentery and the liver surface are particularly important in patients undergoing repeat hepatectomy. In addition, we also considered it important to prevent adhesions on the entire surface of the IVC, around the root of the hepatic vein, and on the surface of the mobilized liver.

There have been reports of the application of adhesion-preventative barriers, as in our study [20]. However, no study to date reported the intentional adhesion-preventative barrier application to the inferior vena cava surface or the root of the hepatic vein. Most of the adhesion-preventative barriers used in previous studies were in sheet form, which hindered their applicability to the hepatoduodenal mesentery and the root of the hepatic vein, areas where the presence of adhesions would significantly hinder manipulations during reoperation. Therefore, the use of AdSpray™, a liquid adhesion-preventative barrier, was useful in liver resection in the present study. Specifically, the complication rate of C-D IIIa or higher was 5.6% in patients treated with AdSpray™, which was not significantly different from that in the control group (5%) in which an adhesion-preventative barrier was not used. These findings were not clearly different from the complication rate of 17% reported by Okubo et al., who used adhesion-preventative barriers in patients undergoing hepatectomy. Altogether, these results support the safety and feasibility of AdSpray™ for MIH.

Although evidence supports that AdSpray™ is safe for use in liver resection, its solution turns into gel mechanism presents challenges because 10 s before gelation, solution tends to flow away from the target area where it is applied, potentially leading to the formation of liquid pools. In their study, Okubo et al. did not find an increase in the rate of complications after AdSpray™ use, although the authors reported a higher rate of intra-abdominal abscesses associated with the spray technique [15]. Similarly, we did not note a significant increase in the rate of complications after AdSpray™ use in the overall cohort. However, as depicted in Figure 6 and Table 4, our subgroup analysis revealed a higher rate of complications in Group 3, which included cases where AdSpray™ was directly applied to the liver transection surface in addition to the standard areas targeted for adhesion prevention, compared to the other groups (14.3% vs. approximately 5%). These results suggest that the use of AdSpray™ to prevent adhesions on the liver transection surface might increase the rate of complications. Okubo et al. postulated that the lower rates of major complication might be attributed to the intentional avoidance of spraying AdSpray™ onto the transection surface, thus preventing the flow of the liquid agent in that area. This speculation implies that early postoperative adhesion formation may contribute to expedited wound healing and highlights the need for a meticulous evaluation of the techniques and types of adhesion-preventative barriers utilized in abdominal surgeries, particularly in procedures where the risk of complications significantly impacts patient outcomes.

The discussion highlights the nuanced approach necessary for the selection and application of adhesion prevention barriers. While these agents offer substantial benefits across various abdominal surgeries, including liver resection, clinicians should carefully balance these advantages against potential risks while considering patient-specific factors and the specifics of the surgical procedure. Moreover, balanced assessment is crucial, especially in surgeries where complications can substantially impact the recovery trajectory of patients.

In conclusion, the present study sheds light on the complexities regarding the use of AdSpray™ in liver resection. While it exhibited a level of safety, its liquid nature presents challenges, which might contribute to an increased rate of complications, particularly in cases where it is applied directly to the transection surface. This prompts a call for the thorough evaluation of adhesion prevention methods, advocating for a cautious approach that considers both benefits and potential risks. Such an approach is vital for optimizing patient outcomes and ensuring successful postoperative recovery.

Finally, we discuss the third finding. MIH has significantly reduced surgical morbidity, mortality, and postoperative hospital stays, emerging as a common procedure for treating liver diseases. In laparoscopic and robotic approaches, CO_2_ remains the preferred choice for pneumoperitoneum; however, CO_2_ embolism remains a notable risk [21,22,23]. Compared to general laparoscopic surgeries, the incidence of CO_2_ embolism in patients undergoing laparoscopic hepatectomy ranges from 1.2% to 4.6%, which is approximately 10 times higher than that reported in standard laparoscopic procedures (0.15%) [24,25,26]. One potential factor contributing to CO_2_ embolism is the pneumoperitoneum management system using AirSeal^®^. This system ensures a clear operative field by constantly evacuating, filtering, and recirculating gas through a dedicated port [27] and maintains optimal insufflation not only by introducing CO_2_ but also by drawing in room air, specifically during sudden pressure drops that occur during suctioning [28,29]. Gas emboli originating from this device should not be considered harmless, as insufflation gas can also contain air. One study reported that the sudden introduction of 200 mL air, roughly equivalent to 1000 mL CO_2_, into the venous system of an adult, was fatal. Therefore, concerns have surfaced regarding the potential risk of air embolism associated with the use of AirSeal^®^ during MIH [30]. Similarly, the dispersion of AdSpray™ throughout the abdominal cavity utilizes a specialized nozzle inserted into the abdomen, raising concerns akin to those associated with AirSeal^®^. Despite these concerns, the intraperitoneal pressure remained consistently low and stable throughout the AdSpray™ application process in the present study. Remarkably, none of the patients experienced any complications, including air embolism, suggesting that the use of AdSpray™ was associated with an exceptionally low risk of air embolism during MIH.

These results offer crucial insights into the safety profile of AdSpray™ in patients undergoing MIH. The absence of air embolism-related complications, even with the use of a specialized nozzle for AdSpray™ application, is reassuring. However, it is imperative to acknowledge the need for further studies to comprehensively evaluate the long-term safety and efficacy of AdSpray™ use in MIH. Thus, continued efforts will be instrumental in refining our understanding of the risks and benefits associated with AdSpray™ with the ultimate goal of optimizing its use and enhancing outcomes in patients undergoing minimally invasive surgery for liver tumors.

Our study findings necessitate careful interpretation, acknowledging several limitations that should be taken into account. First and foremost, the potential for selection bias must be acknowledged, stemming primarily from the considerable difference in patient numbers between the control and case groups—a challenge that was inherent to our study design. Despite this limitation, we diligently pursued a prospective study design, meticulously evaluating 151 consecutive cases. This deliberate approach enabled us to gather a substantial amount of data, enhancing the robustness and depth of our findings.

Additionally, the study was conducted at a single center, limiting the generalizability of our results to broader populations or diverse clinical settings.

Despite these constraints, our study represents a significant step in understanding postoperative abdominal complications in the context of our patient population. The meticulous evaluation of clinical outcomes, coupled with the detailed examination of consecutive cases, provides a solid foundation for our findings. Future research endeavors could benefit from addressing these limitations through multi-center studies, randomized controlled trials, and the incorporation of a wider array of variables for a more comprehensive analysis. These considerations underscore the need for cautious interpretation and the ongoing quest for further insights into optimizing postoperative care strategies.

## 5. Conclusions

The use of adhesion-preventative barriers is critical for the success of repeat hepatectomy in patients undergoing liver resection. The current study used AdSpray™, which differs from other adhesion-preventative barriers by sprayable dextrin hydro-gel based material and verified its suitability for liver resection. Our findings suggest that AdSpray™ is suitable for use during liver resection, given that it can be applied to complex three-dimensional structures. Overall, this study highlights the safety and feasibility of the new spray-type adhesion barrier AdSpray™ in MIH.

## Figures and Tables

**Figure 1 jpm-14-00309-f001:**
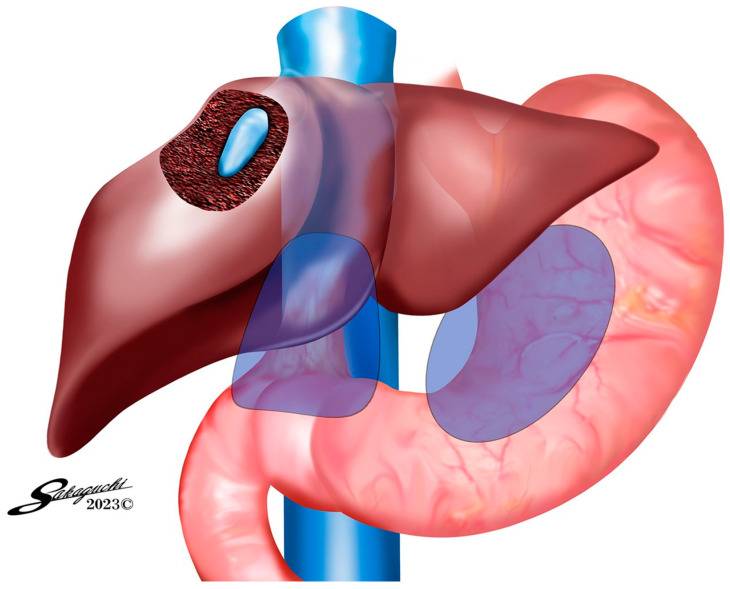
AdSpray™ application areas for Group 1. As indicated by the blue areas, AdSpray™ was applied to the liver surface, gastric surface, hepatoduodenal ligament surface, and port sites. This part was the easiest to use and covered the most cases.

**Figure 2 jpm-14-00309-f002:**
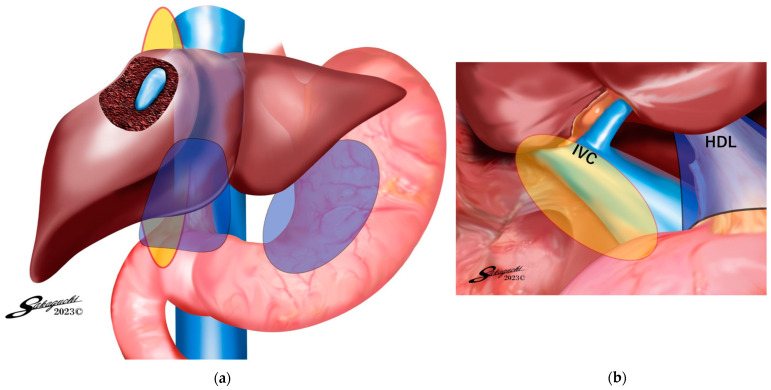
(**a**) AdSpray™ application areas for Group 2. (**b**) Detailed AdSpray™ application areas for Group 2. Areas in which AdSpray™ was applied for Group 2. In addition to Group 1 (the liver surface, gastric surface, hepatoduodenal ligament surface, and port sites), AdSpray™ was applied to the exposed anterior surface of the inferior vena cava, the root of the hepatic vein, and the dorsal surface of the hepatoduodenal ligament. Many of the patients in this group underwent resection beyond segmentectomy. In particular, patients who had undergone a right-sided resection and required observation of the IVC.

**Figure 3 jpm-14-00309-f003:**
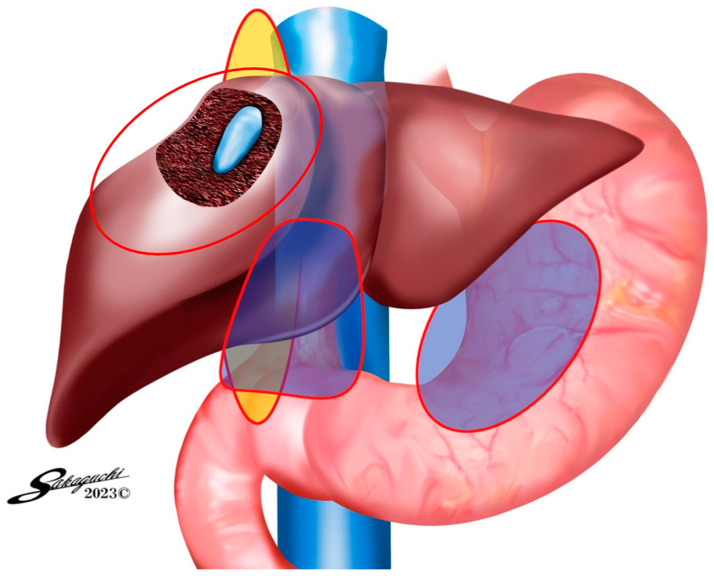
AdSpray^®^ application areas for Group 3.

**Figure 4 jpm-14-00309-f004:**
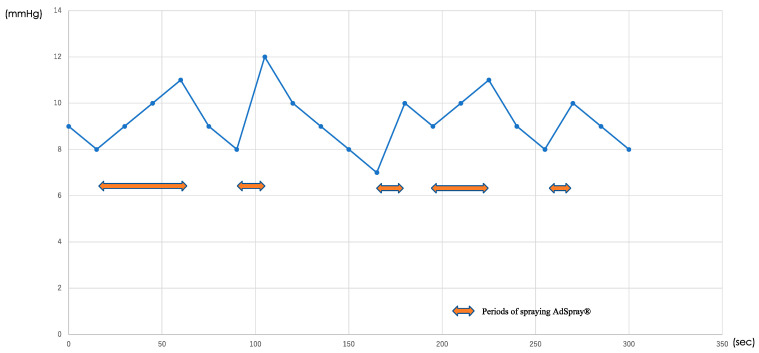
Actual changes in intraperitoneal pressure during AdSpray™ application in one case. Actual changes in intraperitoneal pressure during AdSpray™ application in one case. Double-headed arrows (↔) represent periods of spraying AdSpray™. Although the pressure increased during spraying, it did not exceed 12 mmHg, and no complications occurred.

**Figure 5 jpm-14-00309-f005:**
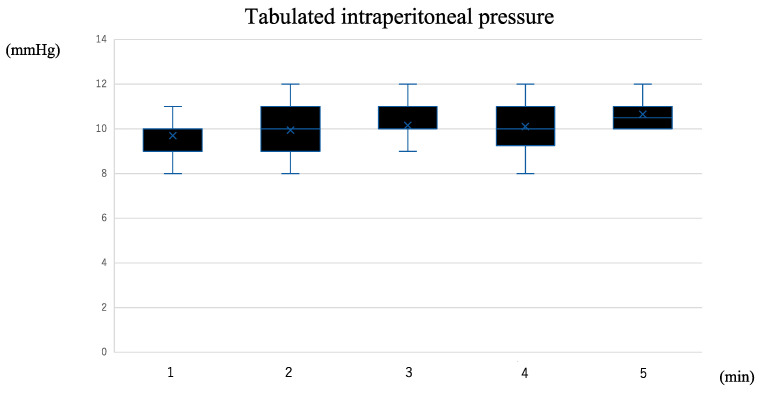
Tabulation of changes in intraperitoneal pressure during AdSpray™ application (n = 20). The box plot tabulates the changes in intraperitoneal pressure for every minute throughout the AdSpray^TM^ application in 20 cases. The median duration of AdSpray^TM^ application was 5 min (range: 2–5 min). The peak pressure during the AdSpray™ application did not exceed 12 mmHg at any time point, and no complications occurred.

**Figure 6 jpm-14-00309-f006:**
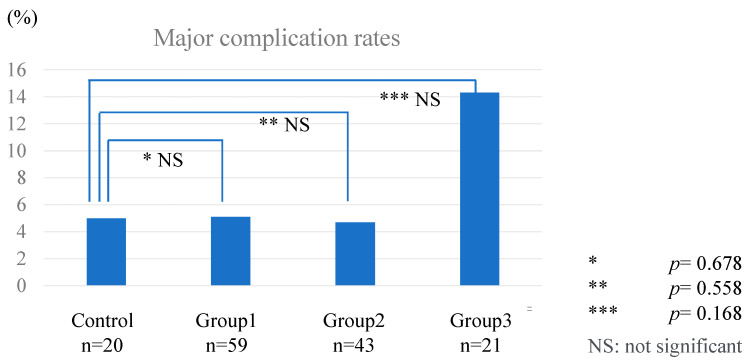
Comparison of major complication rates between Control group, Group 1, Group 2, and Group 3. Major complication rates between subgroups were compared. Although no significant difference was observed, Group 3 tended to show a higher complication rate, which could have been attributed to the application over the resected liver surface. Further investigations are needed to verify the safety of applications over the resected liver surface. NS: not significant.

**Table 1 jpm-14-00309-t001:** AdSpray^®^ applied to areas for the three subgroups.

	1. Liver Surface 2. Hepatoduodenal Ligament (HDL) 3. Port Sites	1. Mobilized Liver Surface 2. Exposed IVC Surface 3. Root of Hepatic Vein	Liver Resected Surface
Group 1 (n = 53)	◯	×	×
Group 2 (n = 49)	◯	◯	×
Group 3 (n = 21)	◯	◯	◯

AdSpray™ application areas for the three subgroups. Patients were divided into three subgroups according to AdSpray™ application areas, as shown. In particular, AdSpray™ was applied to the resected liver surface in Group 3, which remains controversial.

**Table 2 jpm-14-00309-t002:** Background characteristics of patients.

Background Characteristics	AdSpray^®^ Group n = 124	Control Group n = 20	*p*-Value
Age	69 (16–84)	64 (40–84)	0.622
Male gender	76 (61.3%)	12 (60.0%)	0.937
BMI	23.5 (14.8–33.5)	22.5 (14.8–30.8)	0.433
ASA score ≥ 3	0	0	-
Diabetes mellitus	20 (16%)	6 (30%)	0.128
Hepatocellular carcinoma	65 (52.4%)	10 (50%)	0.869
Past history of hepatectomy	45 (36.3%)	8 (40%)	0.564
Albumin (g/dL)	4.0 (2.6–4.8)	4.0 (3.0–4.5)	0.722
Bilirubin (mg/dL)	0.8 (0.3–1.9)	0.75 (0.3–1.0)	0.788
Platelet count (10^4^/mm^3^)	14.7 (1.4–50.6)	18.3 (9.1–34.4)	0.701
Prothrombin (%)	98 (79–124.3)	100 (78–124)	0.701
ICG-R15 (%)	9.1 (2.6–33.5)	1 (1.4–25.9)	0.365
**Surgical Procedures**			
laparoscopic hepatectomy	80 (44.5%)	9 (45.0%)	0.0882
robotic hepatectomy	44 (35.5%)	11 (55.0%)	0.105
≥2 segments resection	11 (8.9%)	2 (10.0%)	0.842
anatomic resection	54 (43.9%)	10 (50%)	0.984
non-anatomic resection	70 (56.1%)	10 (50%)	0.984

Background characteristics of the patients. Comparison of the preoperative background characteristics between the AdSpray™-applied group and control group. No significant difference was observed. Note: The numerical numbers represent the median (range) unless otherwise indicated. Abbreviations: ASA, American Society of Anesthesiologists; BMI, body mass index; ICG-R15, indocyanine green retention rate at 15 min.

**Table 3 jpm-14-00309-t003:** Operative outcomes.

Short-Term Outcomes	AdSpray^®^ Group	Control Group	*p*-Value
	n = 124	n = 20	
Intraperitoneal pressure during spraying AdSpray^®^ (mmHg) * (n = 20)	10 (8–15) *	-	-
Operation time (min)	322 (148–1062)	519 (176–1096)	0.108
Estimated blood loss (g)	65 (5–2432)	174.5 (5–976)	0.132
Major complications	7 (5.6%)	1 (5%)	0.365
Bile leak	5 (4.0%)	0 (0%)	0.168
Abscess	4 (3.2%)	1 (5%)	0.775
Refractory ascites	1 (0.8%)	0 (0%)	0.702
Liver dysfunction	0 (0%)	0 (0%)	-
Bleeding	0 (0%)	0 (0%)	-
Bowel perforation	0 (0%)	1 (5%)	0.365

Comparison of the short-term surgical outcomes between the AdSpray™-applied group and control group. No significant difference was observed. *: Mean and range of intraperitoneal pressure during AdSpray™ application in 20 cases.

**Table 4 jpm-14-00309-t004:** Subgroup analysis of operative outcomes.

Operative Outcomes	Group 1 (n = 59)	Group 2 (n = 43)	Group 3 (n = 21)	Control (n = 20)
Operation time (min)	354 (148–925)	303 (126–1035)	351 (191–732)	519 (176–1096)
Estimated blood loss (g)	65 (2–1279)	64 (5–2432)	62 (3–570)	174.5 (5–976)
Major complication rate	5.1%	4.7%	14.3%	5.0%

## Data Availability

Data are unavailable due to privacy or ethical restrictions.

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
