# Peer review of "The First Prospective Study Investigating the Safety and Feasibility of a Spray-Type Adhesion Barrier (AdSpray™) in Minimally Invasive Hepatectomy: An Analysis of 124 Cases at Our Institution"

_jpm, 2024, doi:10.3390/jpm14030309_

Round 1

Reviewer 1 Report

Comments and Suggestions for Authors

In this study, the authors evaluated the safety and feasibility of a spray-type adhesion barrier 157 (AdSprayTM) in MIH in prospectively registered 124 patients. The application is safe and feasible in a routine clinical setting, and no increase of adverse events could be observed.

Author Response

Thank you for your review. Some text has been added and changed. Please check it again.

Reviewer 2 Report

Comments and Suggestions for Authors

In this prospective study, authors have compared the safety and feasibility of AdSpray in minimally invasive hepatectomy. A total of 124 patients who underwent MIH with AdSpray and 20 controls were analyzed. This is an interesting study and the methodology is well-described, but some issues should be addressed before it is given further consideration.

1. Introduction: More information of hepatectomy and AdSpray should be provided. Currently, the introduction is too short.

2. Table 1: the numbers of patients in each group are suggested to be provided.

3. Figure 4: It is suggested that this figure should be modified and its resolution should be improved. Also, what NS stands for should be mentioned in the figure legend.

4. Proofreading is suggested. Some typos are found. For example, in Line 19, an extra period is found.

Comments on the Quality of English Language

Minor editing of English language required

Author Response

Thank you very much for your review. We have made changes to the main text to address the points you indicated. We would appreciate it if you could check them.

  1. Introduction: More information of hepatectomy and AdSpray should be provided. Currently, the introduction is too short.

I have added information on liver resection and AdSpray to introdution.

  1. Table 1: the numbers of patients in each group are suggested to be provided.

I have added the part you pointed out to the text.

  1. Figure 4: It is suggested that this figure should be modified and its resolution should be improved. Also, what NS stands for should be mentioned in the figure legend.

I have added the part you pointed out to the text.

  1. Proofreading is suggested. Some typos are found. For example, in Line 19, an extra period is fo

I resubmitted the paper for proofreading and confirmed it.

Reviewer 3 Report

Comments and Suggestions for Authors

The present manuscript addresses a significant postoperative problem after surgery, that of surgical adhesions. The research is interesting, however flaws are encountered, mainly deriving from the fact that the control group is way smaller than the case group (20:120).

Regarding format information tends to repeat itself several times in text, Table and Figures; Table 1 contains too much information, it should be shortened for clarity; the Concludions section should be elaborated since it is to evasive and there are not stated the most important findings from the results.

Comments on the Quality of English Language

The quality of English language is acceptable.

Author Response

Thank you very much for your review. I will make changes and additions to the points you indicated and resubmit the manuscript. We would appreciate it if you could check it again.

1. The present manuscript addresses a significant postoperative problem after surgery, that of surgical adhesions. The research is interesting, however flaws are encountered, mainly deriving from the fact that the control group is way smaller than the case group (20:120).

I added my comment as a limitation because it was not possible to add a control group due to the prospective nature of the study.

2. Regarding format information tends to repeat itself several times in text, Table and Figures; Table 1 contains too much information, it should be shortened for clarity;

I have corrected the part you pointed out and submitted it.

3. the Concludions section should be elaborated since it is to evasive and there are not stated the most important findings from the results.

I have corrected the part you pointed out and submitted it.

Round 2

Reviewer 2 Report

Comments and Suggestions for Authors

All issues have been addressed.

Comments on the Quality of English Language

Minor editing of English language required.

Author Response

Thank you very much for your help.
In addition to the changes you suggested, I have increased the number of main manuscript parts  and submitted it again.
We would be grateful if you could evaluate our manuscript.

Reviewer 3 Report

Comments and Suggestions for Authors

The required corrections have been done.

Comments on the Quality of English Language

Minor spelling mistakes 

Author Response

(The authors gave the same response as above.)
